# Mussel periostracum protects against shell dissolution

**Alisha M. Saley** [ORCID][1]*, **Aaron T. Ninokawa**[2], **Abigail Doan**[3], **Brian Gaylord**[1,4]

1 Bodega Marine Laboratory, University of California, Davis, United States of America, 2 Department of Chemistry, State University of New York College of Environmental Science and Forestry, Syracuse, United States of America, 3 Department of Biological Sciences, Santa Rosa Junior College, Santa Rosa, United States of America, 4 Department of Ecology and Evolution, University of California, Davis, United States of America

* amsaley@ucdavis.edu

## Abstract

Reductions to seawater pH challenge the shell integrity of marine calcifiers. Many molluscs have an external organic layer (the periostracum) that limits exposure of underlying shell to the outside environment, which could potentially help combat shell dissolution under corrosive seawater conditions. We tested this hypothesis in adult California mussels, *Mytilus californianus*. We quantified shell dissolution rates as a function of periostracum cover across three levels of reduced pH (7.7, 7.5, and 7.4 on the total scale). Since periostracum can also be eroded over time, we additionally conducted a first-pass examination of whether differing surface textures induced by abrasional processes might influence dissolution rates. We contextualized this set of experiments with measurements of mussel periostracum cover in multiple intertidal habitats. Our results indicate a threefold reduction in shell dissolution rate as periostracum cover increases from 10 to 85% of shell surface area. Dissolution was higher in lower-pH treatments and in treatments where periostracum removal resulted in shells with rougher surface texture, potentially due to increased microtopographic surface area of underlying shell exposed to corrosive seawater. Periostracum loss in the field was greater for mussels at higher shoreline elevations and in sunnier locations, where heat, ultraviolet radiation, and desiccation at low tide may weaken attachment of the periostracum to the shell and. These findings highlight the potential for protective structures of marine organisms to help confront increasingly acute global environmental stressors.

## Introduction

Human-induced perturbations to the seawater carbonate system ('ocean acidification,' or OA; [1]), influence the physiology [2,3], behavior [4,5], and morphology [6,7] of a diversity of marine organisms [8–10]. Impacts then cascade through communities and ecosystems when environmental changes result in disrupted interactions among species [11,12]. Particular attention has focused on OA's capacity to degrade

**Data availability statement:** Dataset Title: Lab incubations of mussels (Mytilus californianus) examining the influence of periostracum cover and pH on external shell dissolution at Marshall Gulch Beach, CA from August 2021 to March 2022 BCO-DMO Metadata Landing page: https://www.bco-dmo.org/dataset/935476 DOI: 10.26008/1912/bco-dmo.935476.1 Citation: Saley, A., Gaylord, B. (2024) Lab incubations of mussels (Mytilus californianus) examining the influence of periostracum cover and pH on external shell dissolution at Marshall Gulch Beach, CA from August 2021 to March 2022. Biological and Chemical Oceanography Data Management Office (BCO-DMO). (Version 1) Version Date 2024-12-28 [if applicable, indicate subset used]. doi:10.26008/1912/bco-dmo.935476.1 [access date] Dataset Title: Lab incubations of mussels (Mytilus californianus) examining the influence of simulated abrasion of periostracum on external shell dissolution at Marshall Gulch Beach, CA from August 2021 to March 2022 BCO-DMO Metadata Landing page: https://www.bco-dmo.org/dataset/935480 DOI: 10.26008/1912/bco-dmo.935480.1 Citation: Saley, A., Gaylord, B. (2024) Lab incubations of mussels (Mytilus californianus) examining the influence of simulated abrasion of periostracum on external shell dissolution at Marshall Gulch Beach, CA from August 2021 to March 2022. Biological and Chemical Oceanography Data Management Office (BCO-DMO). (Version 1) Version Date 2024-12-28 [if applicable, indicate subset used]. doi:10.26008/1912/bco-dmo.935480.1 [access date] Dataset Title: Field measurements of periostracum cover of mussels (Mytilus californianus) from focal population at Marshall Gulch Beach, CA in July and August 2022 BCO-DMO Metadata Landing page: https://www.bco-dmo.org/dataset/935484 DOI: 10.26008/1912/bco-dmo.935484.1 Citation: Saley, A., Gaylord, B. (2024) Field measurements of periostracum cover of mussels (Mytilus californianus) from focal population at Marshall Gulch Beach, CA in July and August 2022. Biological and Chemical Oceanography Data Management Office (BCO-DMO). (Version 1) Version Date 2024-12-26 [if applicable, indicate subset used]. doi:10.26008/1912/bco-dmo.935484.1 [access date].

**Funding:** Award 1: National Science Foundation (NSF); grant # OCE-2129942 (BG); https://

shells of molluscs (see [13] and reviews [14,15]), in part because certain taxa (e.g., mussels) operate as foundation species that create habitat for other organisms by modifying local physical factors, such as flow, temperature, and interstitial chemical conditions [16–18]. Since shells of marine invertebrates protect against environmental extremes and predator attacks, factors that support shell integrity are crucial.

Early OA work [19] conjectured that shells of the deep-sea mussel, *Bathymodiolus septemdierum (formerly B. brevior;* see [20]), persisted under corrosive conditions due to chemical shielding by the organic layer that covers mollusc shells (the periostracum). This layer, common especially in gastropods and bivalves, serves multiple purposes. It provides an initial framework for precipitation of calcium carbonate ($CaCO_3$) [21–24], reduces epibiont fouling [25,26], resists shell abrasion [27], and deters shell-boring predators [28] and parasites [29]. However, although described as a mechanical and chemical 'armor' for molluscs–preventing seawater from contacting the underlying shell via a barrier of chemically hydrophobic layers [30]–its ability to protect against OA-driven shell dissolution remains poorly understood. A few empirical studies have suggested that the periostracum attenuates shell loss in Arctic pteropods [31–33], while others argue that such effects are negligible ([34], with rebuttal in [35]). One study has shown that clay particles attached to the periostracum of an estuarine snail can create an additional low-permeability barrier to acidic seawater, compounding the hydrophobic nature of the periostracum and reducing OA-associated dissolution [36]. Additional research has demonstrated variable periostracum thickness, porosity, and organic composition across mussel species [37–40], characteristics that may affect the degree of shell corrosion by mediating contact between seawater and underlying shell. However, work concerning chemical protections afforded by the periostracum is sparse, and despite informal commentary in the literature (e.g., [41]), we are unaware of additional, direct investigations of the periostracum's capacity to counteract OA effects.

Mussels are one of the most studied calcifier groups in terms of the effects of environmental change, including ocean acidification [42]. Specifically, the California mussel (*Mytilus californianus*) has been instrumental to our understanding of life-stage-dependent shell vulnerability to OA [43–45], and to our knowledge of how individual-level responses to OA may cascade to disrupt relationships among species [46]. The periostracum is non-living tissue and cannot be repaired, which means cumulative damage to it could reduce any putative capacity to protect against corrosive OA conditions external to these animals. Although work on other Mytilid mussels indicates that dissolution of the inner surface of the shell may sometimes be critical (due to the relatively low pH of the extrapallial fluid bathing it; [47]), *M. californianus* secretes a calcite layer of calcium carbonate adjacent to its mantle. This inner calcite layer appears unique to *M. californianus* [48], and is more resistant to dissolution than the aragonitic nacre that characterizes the inner shell of other mussel species. Its presence may thus increase the extent to which shell dissolution in *M. californianus* occurs on external versus inner shell surfaces. Additional mechanisms active in intertidal habitats may also damage the periostracum. *M. californianus* experiences elevated temperatures, ultraviolet radiation, and desiccation at low tide [49–51], as

www.nsf.gov/awardsearch/showAward?AWD_ID=2129942; the funders did not play any role in design, data collection/analysis, preparation of manuscript or publication decisions Award 2: National Science Foundation (NSF) Graduate Research Fellowship; grant # NA (AMS); https://www.nsfgrfp.org/; the funders did not play any role in design, data collection/analysis, preparation of manuscript or publication decisions Award 3: Conchologists of America Grant in Malacology; grant # NA (AMS); https://conchologistsofamerica.org/grants/; the funders did not play any role in design, data collection/analysis, preparation of manuscript or publication decisions Award 4: Russell J. and Dorothy S. Bilinski Fellowship; grant # NA (AMS); https://marinescience.ucdavis.edu/academics/bilinski-fellowships; the funders did not play any role in design, data collection/analysis, preparation of manuscript or publication decisions

**Competing interests:** The authors have declared that no competing interests exist.

well as wave impact [52,53] and abrasion from conspecifics or suspended sediment at high tide [54,55]. Predator and parasite attacks often penetrate and degrade the periostracum [28,55]. This under-recognized organic covering may therefore operate as a nexus for multiple synergistic or antagonistic factors that could influence shell dissolution outcomes under OA.

We conducted laboratory experiments to determine the extent to which percent cover of periostracum on adult *M. californianus* shells influenced rates of loss of external shell, across a range of OA conditions. Multiple physical and biological agents in nature can remove periostracum and simultaneously or subsequently alter the microtopography of the underlying calcium carbonate shell. Therefore, we also explored whether periostracum-denuded shells with differing surface roughness—which could result from natural erosion processes such as sand abrasion—experienced distinct rates of dissolution. We then contextualized results of our experiments by measuring percent cover of periostracum in *M. californianus* at a local field site, where we accounted for relative tidal height and level of sun exposure. The latter two factors affect cycles of heating, drying, and exposure to ultraviolet radiation, which can act as physical agents of periostracum removal and may interact with additional physical and biological agents to accelerate periostracum loss. Our findings reveal a clear role for the periostracum in resisting OA-driven shell dissolution, as well as the potential for interactions with other physical processes that damage this important, but often neglected, organic covering.

## Materials and methods

### Study species

The California mussel, *Mytilus californianus*, inhabits rocky intertidal and subtidal shores of the northeastern Pacific Ocean from southeastern Alaska to southern Baja California [56], forming dense aggregations, or "beds", throughout its range. These beds create complex, three-dimensional structure that provides living space for numerous other species, within which environmental conditions can differ strongly from those outside [16–18]. For the current study, adult mussels (42–64 mm in length) were collected from a single location at Marshall Gulch, California (38.369738 °N, −123.073921 °W) to minimize trait variability driven by site location. Mussels were collected on several low tides spanning August 2021 to March 2022, and transported immediately to the University of California Davis' Bodega Marine Laboratory (< 30 min distance), in Bodega Bay, California. Collections for this study were conducted under annual sport fishing permits issued by the California Department of Fish and Wildlife (Permit Year 2021 and 2022), which authorized the collection of specimens for research purposes. Mussels were held in filtered, flow-through seawater and fed *ad libitum* until used in one of multiple experiments, within 3 weeks.

### Dissolution experiments

Trials examining rates of shell dissolution were conducted in seawater of differing carbonate system conditions, using dead shells of sacrificed mussel individuals. For simplicity, we refer hereafter to seawater carbonate system conditions in terms of pH,

though we emphasize that additional parameters are required to fully describe seawater conditions (S1 Table; see also [57]) and that the thermodynamic tendency for calcium carbonate mineral dissolution is better described by an alternative, correlated parameter (calcium carbonate saturation state; $\Omega$) [58]. We tested effects of the following factors on shell dissolution rates of adult *M. californianus* as three separate experiments: 1) periostracum cover, 2) seawater pH, and 3) shell roughness after periostracum removal.

### Effects of periostracum cover

A first set of experiments determined the relationship between percent surface cover of periostracum and shell dissolution rates under contemporary but chemically stressful seawater conditions characteristic of the California coast. In particular, these conditions correspond to those occurring during strong upwelling events and within shoreline tide pools (pH = 7.5 on total scale, $\Omega_{calcite}$ = 1.0; [5,59,60]). For these and later examinations, the extent of intact periostracum coverage was determined for one of two, symmetrical valves of each *M. californianus* mussel. We photographed individual valves with a 12.2-megapixel digital camera (Google Pixel 4a) and then quantified the area covered by periostracum using ImageJ (software version 1.52a) calibrated to a scale bar (Fig 1). We did not correct for the curved geometry of the shell, which created minor deviations in actual area from the two-dimensional image. Twenty percent of photographs were re-analyzed blind on a later date; these data indicated repeatability to within 3%.

Representative images of *Mytilus californianus* valves used in dissolution experiments, alongside processed images that delineate the surface area of the periostracum (blue) from the underlying shell surface (green). Areas were measured using ImageJ software to calculate surface area coverage for each valve.

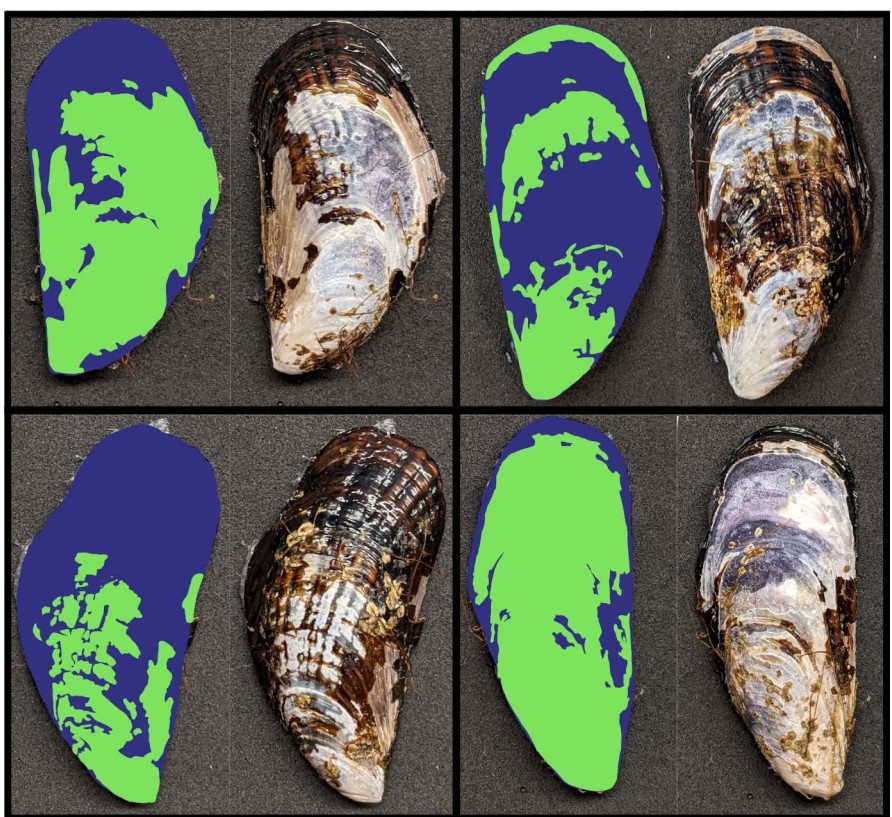

**Fig 1. Shell surface delineation in *Mytilus californianus* valves.**

Following determination of percent cover of periostracum, we quantified shell dissolution rates for a given cover of periostracum using standard alkalinity anomaly techniques [61]. These methods relate calcium carbonate shell loss to the increase in the total alkalinity (TA) concentration of surrounding seawater. Because we were interested in effects of the periostracum in protecting the exterior surface of the shell, we sealed the inner layer of the shell with silicone (Loctite marine silicone sealant) to prevent its contact with seawater. Preliminary trials verified that the silicone did not detectably influence seawater chemistry. For alkalinity anomaly assays and subsequent carbonate system calculations, we recorded seawater properties in 250 mL Nalgene bottles (n = 49), filled without headspace (~300 mL), and recorded temperature, salinity, and pH (Yellow Springs Instruments Professional Plus Sonde) before and at the end of 48 hr. incubations. The two-day incubation period allowed for measurable change in TA but was brief enough to avoid dissolution into deeper, more vulnerable, aragonite shell layers [62]. Values of pH were calibrated to the total scale based on pH spectrophoto-metric measurements of m-cresol dye absorbance at the incubation temperature [63]. Because temperature and salinity affect the dissociation constants of acids and bases—and thus seawater pH—in situ measurements of both parameters are required for accurate pH calibration. Instrument precision in the above pH analyses was approximately 0.01. All incubations were undertaken in a temperature-controlled room in seawater of known salinity (T = 10 °C ± 0.04 SE, S = 34 ± 0.02 SE). Bottles were agitated gently once every 8 hours over the course of the incubations to minimize formation of chemical gradients of their fluid contents, especially any that might form adjacent to the shell surface and thereby impede dissolution processes. We took discrete seawater samples at the onset and end of each incubation for determination of total alkalinity concentration (TA); the samples were frozen for later analysis on a Metrohm 855 Titrosampler. These "before-after" measurements of TA enabled quantification of the increase in alkalinity within each bottle over the duration of the incubation, and thereby the rate of dissolution of calcium carbonate shell material for individual valves of known periostracum cover. During the titration, acid is gradually added to the sample, neutralizing the available buffering ions (i.e., TA). Once buffering capacity is exhausted, a sharp inflection in the pH curve occurs. This inflection point indicates the acid volume required to fully neutralize the sample and is used to calculate TA. While experimental temperature is not needed during TA analysis, but rather, an analysis temperature is specified to construct a precise pH titration curve, it is necessary for calculating the remaining carbonate system parameters. Experimental salinity measurements are critical for both TA determination and subsequent carbonate system calculations. Specifically, salinity affects the ionic strength of seawater and the concentration of major buffering ions; failing to account for it can misrepresent TA values by skewing ion concentration estimates and altering the titration curve. Titration acid concentration was corrected daily using certified ref-erence materials from the laboratory of Dr. Andrew Dickson (Scripps Institute of Oceanography). Samples were analyzed in triplicate (average standard error of triplicates = 1.75 µmol kg$^{-1}$); we selected the median TA value to quantify $CaCO_3$ dissolution rates (µmol $CaCO_3$ hr$^{-1}$ mm$^2$). Dissolution was calculated as the difference in median TA concentration (µmol kg$^{-1}$ seawater) measured before and after incubations, divided by the incubation duration (hours, hr.) and then normalized by shell surface area (mm$^2$). This value was multiplied by the mass of seawater (~300 mL = 0.299 kg) to express results as mass of shell lost, and divided by 2 to account for the conversion of TA to $CaCO_3$, given that 2 TA equivalents correspond to 1 $CaCO_3$ equivalent.

## Effects of pH

Following the first set of experiments, we quantified dissolution rates as a function of periostracum cover across two addi-tional pH levels: a contemporary, but still chemically challenging condition that occurs commonly in our region (pH = 7.7, $\Omega_{calcite}$ = 1.6), as well as a more extreme pH condition that could arise as a synergy between OA and upwelling in future decades (pH = 7.4, $\Omega_{calcite}$ = 0.8). Both of these additional pH levels were selected to focus on seawater conditions antici-pated to be corrosive or borderline corrosive (defined loosely as pH < 7.8), given that dissolution of *M. californianus* shells has been shown to be negligible in pH conditions near the global ocean mean (pH ~ 8.1, total scale) [62]. Such higher pH levels would therefore be mostly uninformative for examining a potential protective role for the periostracum; moreover,

they apply only irregularly at our field site where upwelling of low-pH waters occurs routinely. As in the first set of trials, individual *M. californianus* valves ($n_{pH\,7.7} = 9$, $n_{pH\,7.4} = 16$) were placed into individual incubation bottles, and shell loss was determined using the alkalinity anomaly method based on measured increases in seawater TA described above.

## Effects of shell surface roughness

In a third set of experiments, we performed dissolution assays to gain an understanding of how shell roughness in the absence of periostracum might modify rates of dissolution. Periostracum loss in nature can occur by means of multiple processes, including sand scour from wave-suspended sediments, rubbing of adjacent shells during animal movements, grazing activities of gastropods that live on or within mussel beds, boring organisms that create depressions or drill holes, and endolithic parasites that degrade and erode shells [29,55]. Each resulting type of mechanical damage may alter the microstructural roughness of the shell surface. In our study, we conducted a simple, first-order evaluation of potential consequences of differing surface texture of underlying shell by sanding off the entire periostracum using abrasive paper of two levels of prescribed coarseness (coarse: 270–330 μm nominal diameter [n = 11], and fine: 100–125 μm nominal diameter [n = 12]). In these efforts, we took care to remove as little underlying calcium carbonate as possible. Individual valves were then incubated as before in seawater with a pH level expected in future decades (pH = 7.4).

The difference in dissolution rate normalized by surface area ($\mu mol\ CaCO_3\ hr^{-1}\ mm^{-2}$), was calculated as the difference between the average dissolution rate of control mussel valves (with <5% periostracum cover and no sanding treatment) and the rate of individual valves subjected to either coarse or fine sanding treatments before incubation. Across all three dissolution experiments, control incubations (15% of daily sample size, $n_{control} = 21$ total) of modified seawater were conducted to verify minimal background changes in TA.

## Control of seawater pH

We employed two standard techniques during the three experiments of the study to establish different pH treatments, as equipment used in the first trial was unavailable for the subsequent ones. In the dissolution incubations conducted at pH = 7.5, we modified seawater chemistry using a standard mass-flow control system [64], bubbling gas of a fixed partial pressure of $CO_2$ directly into filtered seawater via flow-through sumps. This process increased the dissolved inorganic carbon (DIC) concentration of the seawater without altering TA, simultaneously reducing its pH. In dissolution trials involving pH = 7.7 and 7.4, we employed an alternative, but equivalent, approach to manipulating seawater pH, due to unavailability of the mass-flow system. In these latter assays, we used direct chemical modification to the seawater carbonate system via equimolar additions of 1 M sodium bicarbonate ($NaHCO_3$) and 1 M hydrochloric acid (HCl). This method (described in detail in [4,65]) duplicates the chemical changes that occur due to direct bubbling of $CO_2$; in particular, it increases DIC and reduces pH but does not change TA. Seawater parameters measured in the laboratory were used to compute the full set of seawater carbonate system parameters employing the *seacarb* package in R (version 3.3.1). In our estimates (S1 Table), we used equilibrium constants determined for similar temperature and salinity (K1 and K2: [66], Kf: [67], Ks: [68]).

## Field assays of periostracum cover

To provide ecological context for the laboratory dissolution experiments, we also measured the percent cover of periostracum of living *M. californianus* in the field, focusing on mussels inhabiting different microhabitats within the intertidal zone. We sampled mussels across two tidal heights and three levels of sun exposure on 17 July and 18 August 2022, from the same population where we had collected mussels for the laboratory trials (Marshall Gulch Beach, California). We sampled from the topmost layer of mussels within a 20 cm x 20 cm quadrat positioned within the center of each of 8 mussel beds at each tidal height (characterized as 'high' or low', −0.3 to +0 m sea level height (SLH) and +0.3 to +1 m SLH), haphazardly choosing the first 20 adult individuals within a pre-selected size range (n = 113 mussels total, 54 mm + 4 mm SD). We characterized sun exposure based on the cardinal direction of the mussel bed orientation, which included mussels oriented

200° – 240° SW (termed 'intense'), 30° – 70° NE (termed 'average'), and 300° – 340° NW (termed 'shade'). We photographed both valves of each mussel, ensuring a scale bar was visible in the field of view for use in ImageJ, and averaged the coverage of periostracum across the two valves as the metric of percent cover for a given individual.

## Statistical analysis

We analyzed the influence of periostracum cover on $CaCO_3$ shell dissolution in *M. californianus* adults for trials at pH = 7.5 using a multiple effects linear model where dissolution rate was normalized to shell surface area ($mm^2$) to control for variations in shell size. Normalized dissolution rates were square root transformed to adhere to linearity requirements for normally distributed residuals in linear model assumptions. We used backward stepwise model selection using F test significance as the criteria for omitting terms, resulting in a final model that employed both percent periostracum cover and valve length as fixed predictors. Throughout model selection tests, trial date did not significantly explain variability as a random effect, so was omitted. Model residual homoscedasticity was verified using a Breusch-Pagan test (BP = 1.0525, df = 2, p-value = 0.5908), and assumptions of linearity and additivity were assessed visually using model residuals and QQ-plots.

Because in our second experiment we were also interested in how seawater pH independently affects patterns of dissolution as a function of periostracum cover, we used a multiple effects linear model across the two additional pH levels (pH = 7.4, 7.7). Here, we included pH as a numerical fixed predictor. Due to a constrained sampling size and narrow range of shell lengths (52 mm ± 4 mm SD), shell length was not included as a fixed predictor in this model in order to conserve degrees of freedom. Normalized $CaCO_3$ dissolution rate was again square root transformed, and backward stepwise model selection was used to estimate the significance of fixed predictors and omit the non-significant interaction between periostracum cover and pH. A Breusch-Pagan test confirmed residual homoscedasticity (BP = 3.5024, df = 3, p-value = 0.3204), and other assumptions were assessed visually as before.

To understand results of the third experiment examining whether shell roughness arising from abrasion with grit of differing coarseness might influence dissolution in periostracum-free shells, we computed the difference between dissolution rates of mussel valves in abrasion treatments, and those of unsanded control valves (collected from the field with < 5% periostracum cover; i.e., 'denuded') exposed to similar low-pH conditions. We then used Welch's independent sample t-test to determine whether differences in average dissolution existed between the sanding treatments. Assumptions of normality and homogeneity of variances were assessed visually using QQ-plots.

For the field contextualization component of the study, we used a two-way ANOVA to compare average periostracum cover in adult mussels found at two relative tidal heights and three levels of sun exposure. We omitted the interaction term between the treatments after confirming it failed to explain significant variability in the model. Using tidal height and sun exposure as categorical predictors to explain periostracum cover, we then ran a TukeyHSD test (95% confidence) to identify which microhabitat types differed from one another via post-hoc, pairwise comparisons.

## Results

### Periostracum cover and shell loss

Normalized $CaCO_3$ dissolution rates (i.e., dissolution rate per shell area) declined strongly with higher percent cover of periostracum in adult California mussels (Fig 2; Table 1). Specifically, in treatment conditions of pH = 7.5 (mean = 7.51 ± 0.01 SD; $\Omega_{calcite}$ = 1.0), valves with less than 15% cover exhibited normalized dissolution rates three times higher than those with 85% periostracum cover. Normalized dissolution rates also displayed an independent, positive relationship with shell length (Table 1), a metric often associated with age. This latter trend, while weaker than the one tied to periostracum cover, suggests that older shells may have higher per-area dissolution rates than younger ones (Fig 3), perhaps due to accumulated cracks or small holes, including imperfections that are not readily visible in images used to assay periostracum cover.

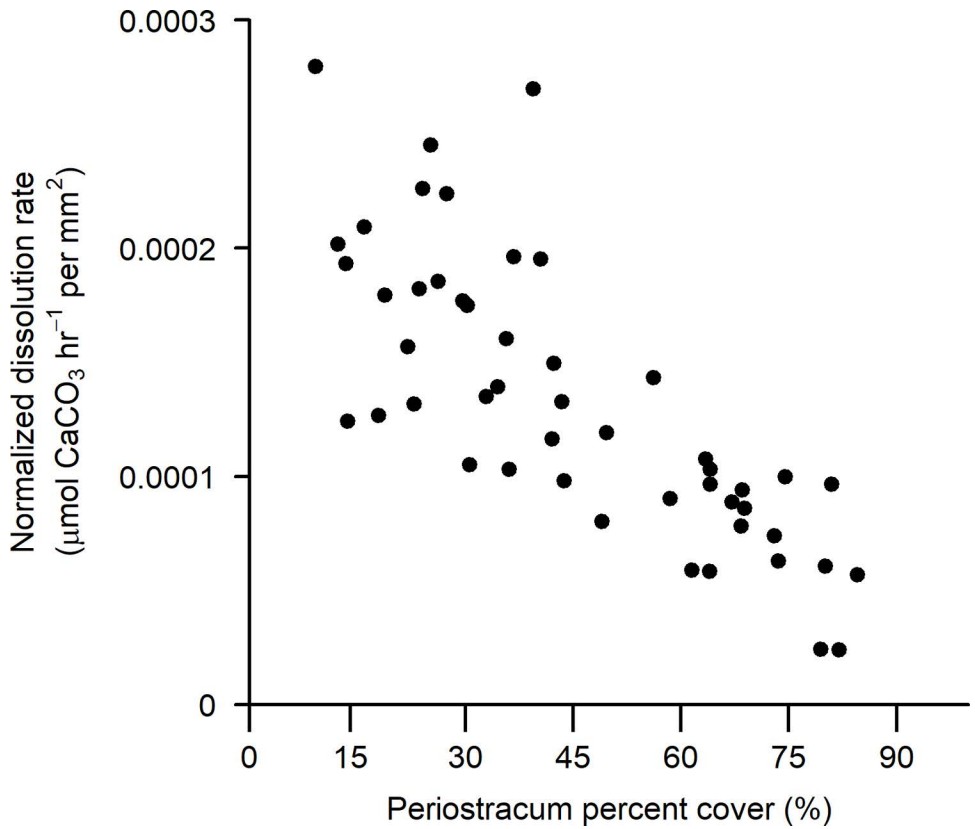

**Fig 2. Dissolution rate of *Mytilus californianus* valves as a function of periostracum surface area cover in low pH seawater.** Dissolution rate normalized by shell surface area (μmol CaCO₃ hr⁻¹ per mm²). of adult mussel valves as a function of percent cover of periostracum, measured after exposure to low pH seawater conditions (pH = 7.5, total scale) for approximately 48 hrs.

### Effects of seawater pH

Further reductions in seawater pH induced an additional, independent increase in normalized dissolution rate, beyond the effects of periostracum cover (Fig 4; Table 2). Rates of shell loss doubled as seawater pH decreased from 7.7 (mean = 7.74 ± 0.01 SD; $\Omega_{calcite}$ = 1.6) to 7.4 (mean = 7.38 ± 0.01 SD; $\Omega_{calcite}$ = 0.8). No statistical interaction between pH and periostracum cover was apparent.

**Table 1. Results of multiple fixed-effects model testing the independent influence of periostracum cover (%) and shell length (mm) on normalized dissolution rates (CaCO₃ dissolution rate per shell area) in adult *Mytilus californianus* valves.**

| Fixed Effects (multiple $r^2$ = 0.6937, adjusted $r^2$ = 0.6804) | Estimate | Std. Error | t-value | df | F-statistic | p-value |
|---|---|---|---|---|---|---|
| Intercept† | 6.44E-03 | 3.79E-03 | 1.697 | | | 0.0965 |
| Periostracum cover (%) | −9.12E-05 | 1.05E-05 | −8.721 | 46 | 76.064 | **< 0.0001** |
| Shell length (mm) | 1.60E-04 | 6.51E-04 | 2.464 | 46 | 6.0713 | **0.0175** |

F-statistics and p-values were recorded during backward stepwise model selection. Bolded values denote a significant effect (alpha < 0.05). The final model, sqrt(normalized dissolution rate) ~ periostracum cover + shell length, accounted for 68% of the total variation.

†Intercept represents 0% periostracum cover and 0 mm shell length

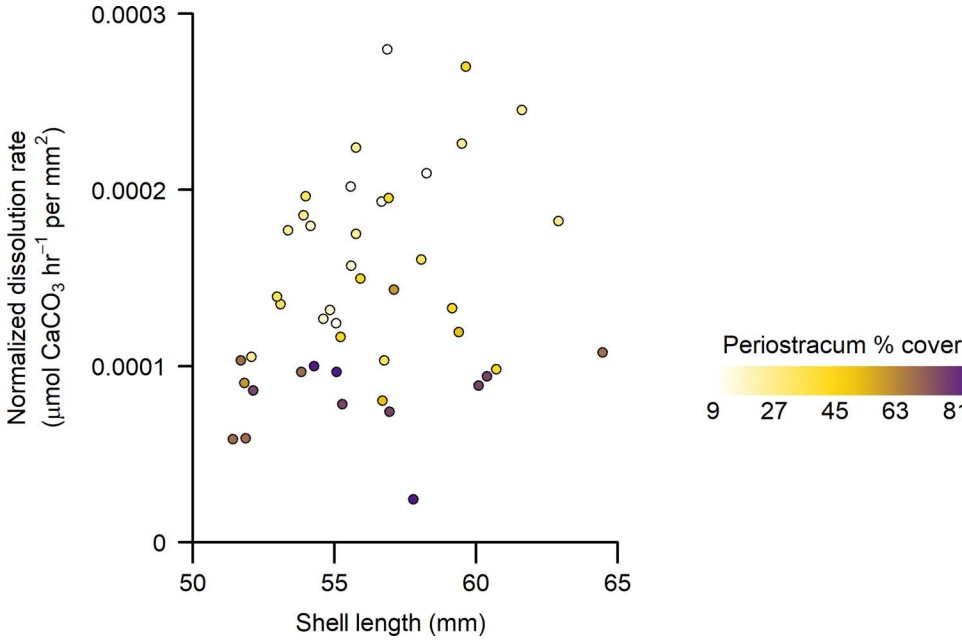

**Fig 3. Dissolution rate of *Mytilus californianus* valves as a function of shell length and periostracum surface area cover in low pH seawater.**
Dissolution rate normalized by shell surface area (µmol CaCO$_3$ hr$^{-1}$ per mm$^2$) of adult mussel valves as a function of shell length (mm) and percent cover of periostracum. Shell loss was measured after approximately 48 hrs. of exposure to low pH seawater conditions (pH = 7.5, total scale).

## Shell surface roughness

Rates of normalized shell dissolution in the absence of periostracum were influenced by the coarseness of the abrasive grit used to remove the periostracum (Table 3). In trials of seawater pH = 7.4, normalized dissolution rates of periostracum-free shells when coarse grit was used as an abrasive agent were elevated relative to those of unsanded control shells that were collected lacking periostracum from the field (Fig 5). In contrast, when fine grit was used to remove the periostracum, resulting in a more polished shell surface, normalized dissolution rates decreased relative to those of periostracum-free control shells (Fig 5).

## Field measurements of periostracum cover

Mussels from locations lower on the shore had higher periostracum cover than conspecifics found at higher tidal heights (Table 4), regardless of microhabitat (sunny versus shadier exposure; see dark vs. light boxes in Fig 6). In addition, mussel shells from shadier locations showed increased cover of periostracum compared to mussels with either moderate or intense sun exposure (Fig 6).

The final model, periostracum cover (%) ~ sun exposure + tidal height, suggests that higher periostracum cover is associated with shady locations positioned lower on the shore. A Tukey HSD test was used to compare the levels within each fixed predictor, indicating higher periostracum cover from shadier beds. Bolded values denote a significant effect (alpha < 0.05).

## Discussion

The extent of periostracum coverage in adult California mussels determined the rate of external CaCO$_3$ shell dissolution under corrosive seawater conditions. This effect was exacerbated in seawater of lower pH. Shells of larger and

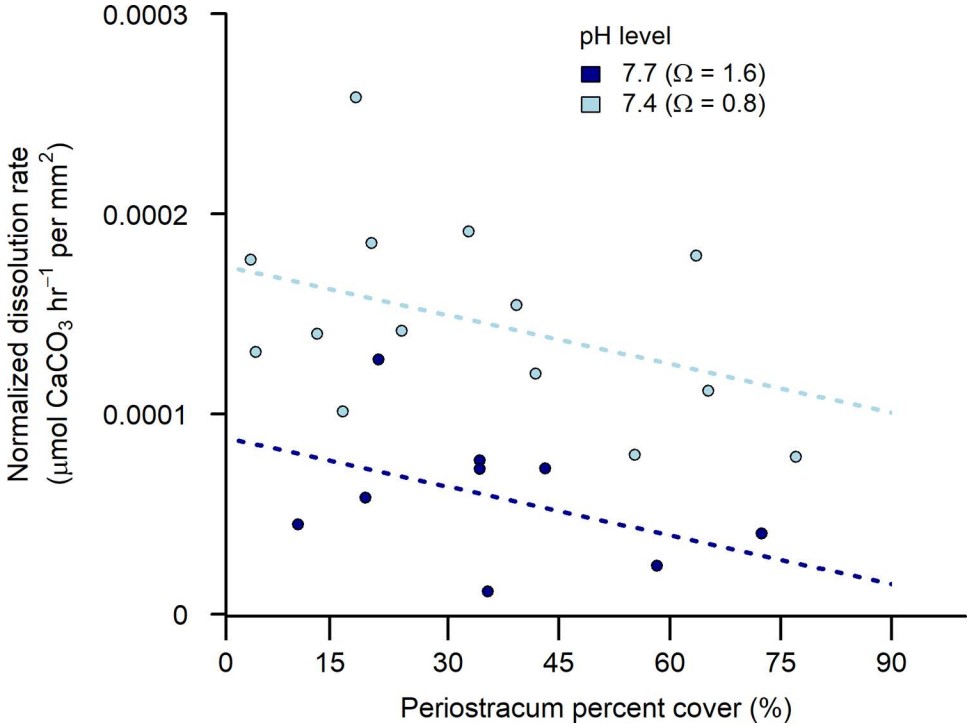

**Fig 4. Dissolution rate of *Mytilus californianus* valves as a function of the dual parameters of periostracum surface area cover and low pH seawater.** Dissolution rate normalized by shell surface area (µmol CaCO₃ hr⁻¹ per mm²) of adult mussel valves as a function of two additional low pH levels of 7.4 and 7.7 (total scale), and percent cover of periostracum. Dashed lines represent dissolution rate predictions from the final model (Table 2). pH levels are reported alongside the saturation state ($\Omega$) for calcite mineral forms of calcium carbonate.

**Table 2. Results of a multiple fixed-effects linear model testing the independent effects of periostracum cover and pH on normalized CaCO₃ dissolution rates (per shell area) in adult *Mytilus californianus* valves.**

| Fixed Effects (multiple $r^2$ = 0.5667, adjusted $r^2$ = 0.5482) | Estimate | Std. Error | t-value | df | F-statistic | p-value |
|---|---|---|---|---|---|---|
| Intercept† | 1.25E-01 | 2.41E-02 | 5.167 | | | **< 0.0001** |
| Periostracum cover (%) | −5.33E-05 | 2.52E-05 | −2.116 | 22 | 4.4795 | **0.0459** |
| pH (total scale) | −1.49E-02 | 3.22E-03 | −4.638 | 22 | 21.514 | **0.0001** |

F-statistics and p-values were recorded during backward stepwise model selection. Bolded values denote a significant effect (alpha < 0.05). The final model, sqrt(normalized dissolution rate) ~ periostracum cover + pH (total scale) accounted for 55% of the total variation.

†Intercept taken at 0% periostracum cover, 0 pH and 0 mm shell length

presumably older individuals were more vulnerable to external dissolution. Our work also suggests that shell dissolution is enhanced if erosion of the periostracum occurs by means of processes that result in a rougher shell surface. The latter trend could derive from increases in surface area at microtopographic scales, although further studies are required to confirm this hypothesis. Field assays suggest that periostracum of *M. californianus* may be preferentially lost in sunnier locations and at elevated locations on the shore, where cycles of heating, drying, and exposure to ultraviolet light can be expected to be more pronounced.

Our work builds on broader functional examinations of the periostracum in molluscs. Prior research details the contribution of this organic surface layer to initial shell deposition [21,23] and to repairing shell damage along the growing edge

**Table 3. Results of Welch's individual t-test comparing differences in normalized shell dissolution between valves abraded with sandpaper (fine or coarse grit), and normalized CaCO$_3$ dissolution in unabraded control valves also lacking periostracum[†].**

| Predictor | Levels | Level means | 95% CI | t-value | df | p-value |
|---|---|---|---|---|---|---|
| Sandpaper Grit | Coarse | 1.12E-04 | | | | |
| | vs. | | 1.96E-04 – 2.83E-04 | 11.703 | 16.46 | **< 0.0001** |
| | Fine | −1.28E-04 | | | | |

Bolded values denote a significant effect (alpha < 0.05). The final model, difference in normalized dissolution ~ sanding treatment, suggests that dissolution rates are influenced by the coarseness of agents that abrade the periostracum.

[†]H$_0$: difference in means between sanding groups = 0

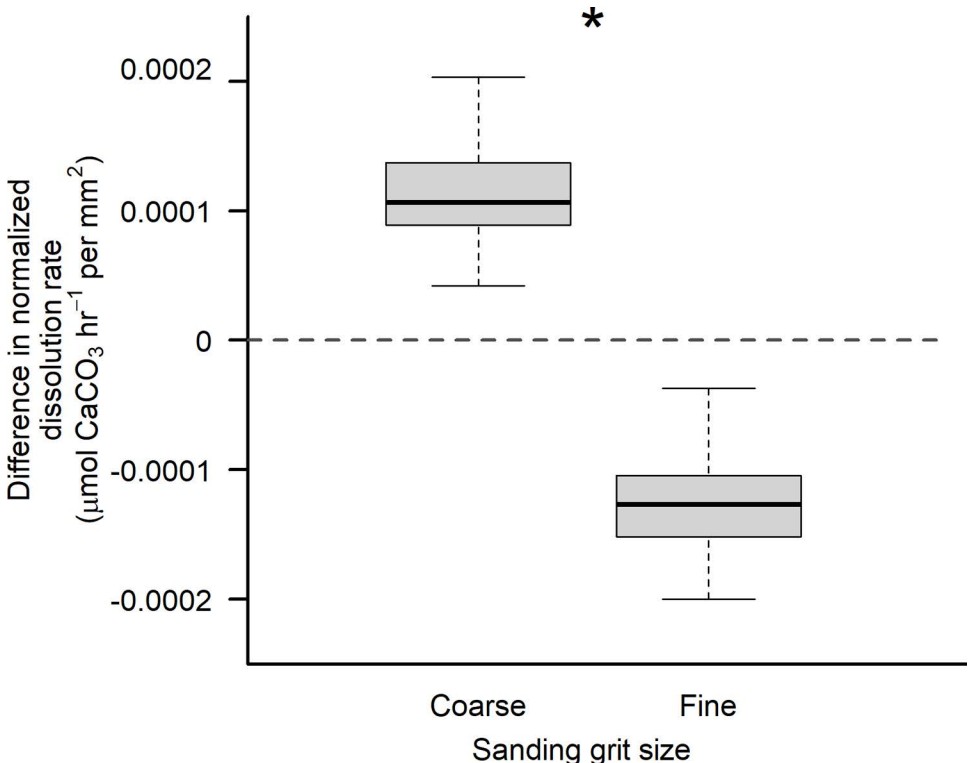

**Fig 5. Difference in dissolution rate between abraded and unabraded *Mytilus californianus* valves as a function of abrasive grit coarseness.** Difference in dissolution rate normalized by surface area (μmol CaCO$_3$ hr$^{-1}$ per mm$^2$) between abraded mussel valves and unabraded control valves (dashed line), when all shells lack periostracum, as a function of abrasive grit coarseness. Note that a positive dissolution rate indicates dissolution greater than that of the controls, while a negative dissolution rate indicates shell loss rates lower than those of the controls. Trials were conducted in seawater of pH = 7.4 (total scale). Note that a change of 0.001 μmol CaCO$_3$ hr$^{-1}$ per mm$^2$ represents a shift of 45% relative to controls.

[69] by providing a corrosion-resistant and minimally porous scaffolding for shell deposition. Other studies have examined anti-fouling properties of periostracum [26] and its relationship to boring parasites [25,29]. In addition to work evaluating the role of the periostracum in protecting pteropod shells against chemical corrosion [31,35], recent research has explored potential effects of periostracum loss on body temperature, noting the periostracum's often-dark color and capacity to increase solar heating [55]. However, the full extent to which the periostracum of other molluscs mitigates effects of environmental change remains unclear and warrants further investigation. Our work suggests a potential benefit of the

**Table 4. Results of two-way ANOVA model examining the effects of sun exposure and relative tidal height on measured periostracum cover of adult *Mytilus californianus* valves in the field.**

| Source of Variation | Sum of Squares | df | Mean Square | F-value | p-value |
|---|---|---|---|---|---|
| Sun Exposure | 3413 | 2 | 1706 | 5.1277 | **0.0074** |
| Shoreline Height | 15638 | 1 | 15638 | 46.9967 | **< 0.0001** |
| Residuals | 36270 | 109 | 333 | | |
| **Tukey HSD comparison** | **Difference** | **Std. Error** | **t-value** | **p-adj** | |
| *Sun exposure* | | | | | |
| Average vs. Intense | −1.750 | 4.213 | −0.415 | 0.9706 | |
| Shade vs. Intense | 10.485 | 4.185 | 2.505 | **0.0493** | |
| Shade vs. Average | 12.235 | 4.213 | 2.904 | **0.0166** | |
| *Shoreline height* | | | | | |
| Low vs High | 23.553 | 3.436 | 6.855 | **< 0.0001** | |

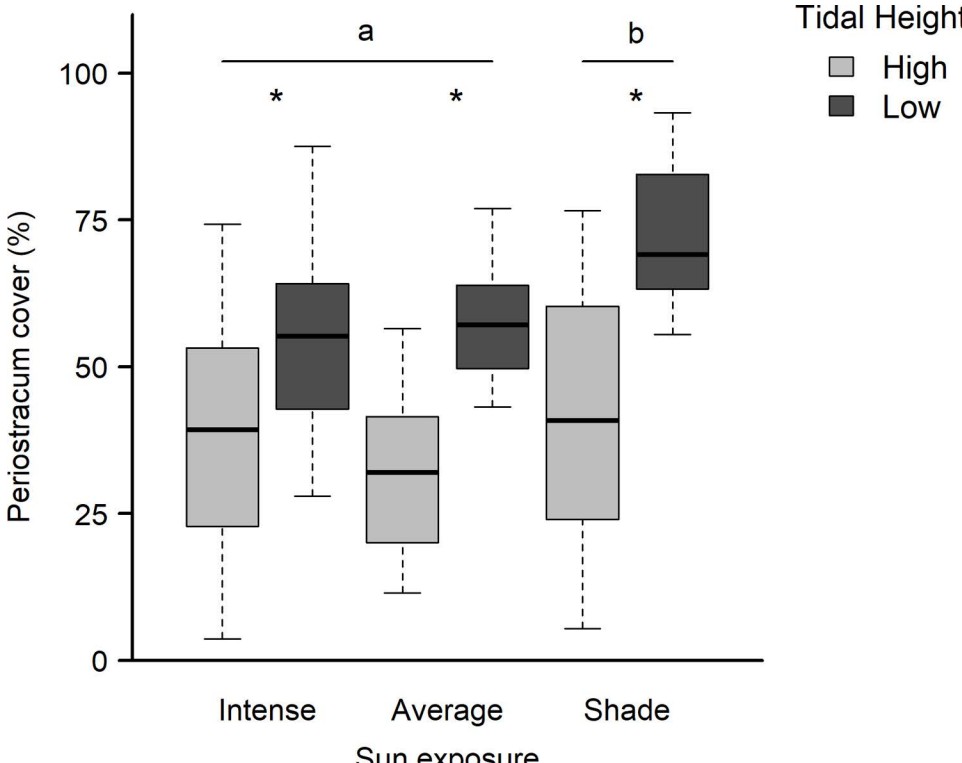

**Fig 6. Measured percent periostracum of adult *Mytilus californianus* in the field as a function of sun exposure (x-axis) and relative shoreline height (gray scale).** A total of n = 113 individuals were sampled across 8 mussel beds on 17 July and 18 August 2022, at Marshall Gulch, California. A two-way ANOVA and post-hoc Tukey HSD test revealed significantly higher cover on individuals positioned higher on the shoreline (*) or living on shady rock surfaces (a vs. b).

periostracum in helping *M. californianus* mussels cope with ocean acidification, a process likely to influence the distributional pattern of other molluscs in future decades.

Results of our experiments show a linear increase in dissolution rate as the percent cover of periostracum declines, under conditions of reduced pH. Areas of the shell covered by the periostracum were likely protected from dissolution

due to its low porosity, unlike exposed areas of the underlying shell, which are more vulnerable to dissolution. However, our study is less informative for understanding how periostracum cover might interact with pH over a more expansive pH range or with periodic exposure to corrosive conditions. For example, prior work has demonstrated non-linear trends in rates of shell dissolution across wider ranges of pH [62,70], which is likely to occur in coastal habitats with significant carbonate system variability. We also note that processes governing periostracum loss can be complicated. Damaged periostracum, which can result from physical processes and species interactions, allows seawater to penetrate beneath it to contact underlying shell, as initial spots of degradation facilitate further detachment and exacerbated dissolution, especially when coupled with episodes of wetting/drying, wave action, and abrasion from sediments [31,71,72]. Multiple drivers of periostracum removal may also operate in concert. For example, hydrodynamically energetic sites along exposed coastlines characterized by larger waves tend to have coarser sand grain sizes [73], exposing mussel inhabitants to riskier forms of abrasion and larger flow forces. The latter could enhance shear-induced removal of weakly attached or flaking periostracum. On the other hand, hydrodynamically energetic sites often experience vigorous wave spray and concomitant moisture-associated cooling, which could attenuate thermal damage. Additionally, predation and parasitism may interact with, and be influenced by, physical damage or intertidal zone characteristics, potentially leading to non-additive effects on periostracum loss. While these complexities were beyond the scope of the present study, disentangling the effects of hydrodynamic setting and species interactions on mussel periostracum integrity remains a valuable direction for future research.

Our findings reveal an independent relationship between normalized dissolution rate and shell length. This pattern, as noted above, could be related to shell age. Longer, larger shells of older individuals are often weathered, and such weathering might introduce small cracks or perforations into the periostracum. We did not observe such forms of damage in our photographic images, but it is not unreasonable to expect that such types of periostracum degradation would be detectable only under microscopy [35]. Another hypothesis could be that older and more heavily weathered shells provide seawater access to more soluble mineral forms that appear deeper in the shell. In *M. californianus* specifically, the outer prismatic layer of the shell is composed of calcite, which is less soluble than the deeper, interior aragonite shell layer [48]. It similarly appears that shell layers produced earlier in life by oysters are more vulnerable to dissolution [74]. If more susceptible portions of shells become increasingly exposed to exterior seawater as outer components accumulate damage, one might expect not only greater vulnerability to OA in older mussels, but also a progressive acceleration of dissolution in a given mussel individual as they age. Although work suggests that mussel shell repair following damage (e.g., from penetrating bioeroders or predation attempts) can proceed effectively under OA conditions [75], further research into the adaptive capacity to respond to such damage would be of value.

A comprehensive understanding of shell vulnerability in future seawater conditions must consider how molluscs will respond through plasticity (acclimatization) and/or genetic adaptation in chemically heterogeneous environments. For example, some mussel populations deposit thicker periostracum that adheres more tightly to the shell [19,39], likely by slowing down the 'conveyor-belt' shell precipitation process [23]. It is suggested that exposure to variable seawater pH and salinity [40], food supply [76], length of the individual [39] and increased predation pressure [77] influence the degree of periostracum thickening within an individual mollusc shell, and among individuals from a population. We did not measure differences in periostracum thickness within and among individuals in our study, but future work exploring the basis for such variation and its influence on dissolution vulnerability in *M. californianus* would be beneficial, given the known plasticity of this trait. Likewise, efforts to understand ecological trade-offs for periostracum strength would be advantageous. Improved periostracum adhesion and thickening has primarily been observed in bivalve species living in chemically stressful habitats, suggesting that individuals may invest more to improve the quality of the periostracum to counter potential dissolution stress on the shell [19,39,78]. That said, morphological changes usually come at an energetic cost, and are dictated by energetic state (which can in turn be affected by other processes such as reproduction), which could result in trade-offs to other forms of growth (see [79]). Alternatively, when food is abundant, bivalves may upregulate

energetically costly processes to internally thicken the calcium carbonate shell to compensate for surface dissolution [14,35,80]. When food becomes limiting, mussels may produce thinner periostracum [39], which could result in more damage and dissolution of shells, and elevated predation risk. In general, the costs of producing periostracum of differing structural traits are poorly described, as are the costs of sublethal damage associated with chipping, flaking, or hole-boring by predators, which often damage periostracum.

As seawater pH and calcium carbonate saturation state continue to decline locally and globally, we will likely see dissolution occurring alongside disruptions to other fundamental aspects of mussel biology. For example, we may see weakened byssus [81–83] or immune function [84] accompanying amplified dissolution rates and changes to the periostracum itself, under OA [40,85,86], which could reduce the energy available for shell maintenance and repair (but see [75]). Joint effects could elevate mortality rates beyond the effects of each driver alone. Further, low-tide heat exposure may impose increased stress on *M. californianus* under predicted temperature increases under climate change scenarios [50]. We observed that shadier and lower-elevation locations provided better protection against periostracum loss for *M. californianus* mussels, and such microhabitats might become thermal refuges as air temperatures rise. Other factors might operate oppositely. Endolithic parasites and predation attempts that remove the periostracum from mussel shells change them from brown to white, decreasing heat loads. Studies that account for interactions among multiple axes of biological stress are vital and should be prioritized.

Regardless of such complexities, our work demonstrates that the periostracum can protect against external shell dissolution, potentially helping combat the corrosive action of ocean acidification. This finding highlights the need to consider this organic protein layer as a resistive shield against OA-derived shell loss, while recognizing that this shield is vulnerable to damage and loss from a spectrum of other environmental stressors.

## Supporting information

**S1 Table. Description of the full set of carbonate system conditions for each of the laboratory experiments.** We measured total alkalinity and pH in order to estimate the remaining seawater parameters, along with salinity and temperature. We report calcium carbonate saturation state values for the both aragonite and calcite mineral forms but note the higher presence of aragonite mineral in *Mytilus* shells relative to the calcite mineral form.
(DOCX)

## Acknowledgments

The Marshall Gulch region was home to Coastal Miwok and Southern Pomo Indigenous tribes (organized now as the Federated Indians of Graton Rancheria), whose diet included a variety of coastal marine species, including mussels.

## Author contributions

**Conceptualization:** Alisha M Saley, Aaron T Ninokawa, Brian Gaylord.

**Data curation:** Alisha M Saley, Abigail Doan.

**Formal analysis:** Alisha M Saley.

**Funding acquisition:** Alisha M Saley, Brian Gaylord.

**Investigation:** Alisha M Saley.

**Methodology:** Alisha M Saley, Aaron T Ninokawa, Abigail Doan.

**Project administration:** Alisha M Saley.

**Resources:** Alisha M Saley, Brian Gaylord.

**Supervision:** Alisha M Saley, Brian Gaylord.

**Validation:** Alisha M Saley.

**Visualization:** Alisha M Saley.

**Writing – original draft:** Alisha M Saley.

**Writing – review & editing:** Alisha M Saley, Aaron T Ninokawa, Abigail Doan, Brian Gaylord.

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
