## [Decision Letter · Decision Letter 0]

Dear Dr. Saley,

Thank you for submitting your manuscript to PLOS ONE. After careful consideration, we feel that it has merit but does not fully meet PLOS ONE’s publication criteria as it currently stands. Therefore, we invite you to submit a revised version of the manuscript that comprehensively addresses the points raised during the review process.

We look forward to receiving your revised manuscript.

Kind regards,

Michael Schubert

Academic Editor

PLOS ONE

3. Thank you for stating the following in the Acknowledgments Section of your manuscript: “The Marshall Gulch region was home to Coastal Miwok and Southern Pomo Indigenous tribes (organized now as the Federated Indians of Graton Rancheria), whose diet included a variety of coastal marine species, including mussels. This work was funded by National Science Foundation (NSF) grant OCE-2129942. AMS is also grateful for an NSF Graduate Research Fellowship and for awards from Conchologists of America Grant in Malacology and a Russell J. and Dorothy S. Bilinski Fellowship.”

Please remove any funding-related text from the manuscript and let us know how you would like to update your Funding Statement. Currently, your Funding Statement reads as follows: “Award 1: National Science Foundation (NSF); grant # OCE-2129942 (BG); https://www.nsf.gov/awardsearch/showAward?AWD_ID=2129942; the funders did not play any role in design, data collection/analysis, preparation of manuscript or publication decisions”

4. Thank you for uploading your study's underlying data set. Unfortunately, the repository you have noted in your Data Availability statement does not qualify as an acceptable data repository according to PLOS's standards.

Reviewers' comments:

Reviewer's Responses to Questions

**Comments to the Author**

1. Is the manuscript technically sound, and do the data support the conclusions?

Reviewer #1: Yes

Reviewer #2: Yes

2. Has the statistical analysis been performed appropriately and rigorously?

Reviewer #1: Yes

Reviewer #2: Yes

3. Have the authors made all data underlying the findings in their manuscript fully available?

Reviewer #1: No

Reviewer #2: Yes

4. Is the manuscript presented in an intelligible fashion and written in standard English?

Reviewer #1: Yes

Reviewer #2: Yes

Reviewer #1: In this study, the authors provided evidence that the periostracum helps mussels protect against shell dissolution. I find this paper to be an interesting addition to the understanding of the protective abilities of bivalves against ocean acidification and believe it is publishable in PLOS ONE. However, the paper requires revision due to several uncertainties that need clarification before publication. Specifically, it was difficult to determine how the periostracum cover influenced the dissolution rate. I would recommend including sample images of the periostracum and the dissolved surfaces of mussels.

• L139: Which of the two valves was used? Depending on the species, the two valves may not be symmetrical.

• L141: Is it possible to see the periostracum with the naked eye? What criteria were used to identify the periostracum?

• L152-153: Please provide additional details in the supplementary material.

• Figure 1: It was difficult to understand how varying percentages of periostracum cover influenced the dissolution rate. Could you provide sample images for each end of the spectrum? This would make it much clearer.

• Figure 4: Why do mussels abraded with fine sand show a negative dissolution rate (lower than controls)?

• L435: Is there a specific reason for not using SEM? SEM offers much higher resolution and could provide more precise results.

• L456: Why wasn’t the periostracum thickness measured, even though it is important?

Reviewer #2: This study has the objective to investigate the impact of periostracum coverage on the ability of California mussels to resist shell dissolution under a diverse environment stressors, especially ocean acidification, where the authors explored how shell size, age and environmental conditions (such as food availability and wave exposure) can interact with the periostracum to influence dissolution rates. The paper is very well written, well-organized and insightful, providing clear explanations of the experimental design and results.

Abstract

Line 26 - You can change “possibility” to “hypothesis”.

Line 29 - You can change “Because” to “Since”.

Line 36 - If you have enough space, you should specify the meaning of “rougher surface texture” and its effects on dissolution.

Line 38 to 39 - How the desiccation at low tide can contribute to damage?

Introduction

Line 50 - Add how mussels modify local physical factors and its impacts (water flow, sediment stability, etc).

Line 60 - On mechanical and chemical armour, does is mean periostracum will prevent shell thinning? Or does it neutralize acidic conditions?

Line 64 - Why particles attached reduces OA-associated dissolution?

Line 67 - What are the implications?

Line 69 to 70 - Are there variations in the chemical compositions among different mussels species that can affect its ability to withstand OA?

Line 75 - You can change “may propagate” to “may disrupt”.

Line 76 - Change “not living tissue” to “non-living tissue”.

Line 78 to 79 - What are the molecular mechanisms in response to the acidification? Microbial activity will impact or influence the degradation of periostracum?

Line 80 to 81 - How periostracum degradation vary across different intertidal zones/ecossystems?

Line 84 - Does UV radiation has any impact/role on structural integrity?

Line 92 - Change “undertook” to “conduct”.

Line 92 to 94 - Is it possible to identify the specific conditions were the periostracum loss accelerates?

Line 94 to 96 - Predation or microbial actions are most effective in removing periostracum?

Line 98 to 100 - What are the long-term effects of tidal exposure?

Methods

Study species

Capitalize the word “species” -> Species.

Line 110 to 111 - Does the environmental factors in these regions have any influence in their reproductive patterns?

Line 113 to 114 - What are the key factors that can influence the diversity and survival of the species?

Line 117 - Add a coma after 2022.

Dissolution experiments

- Capitalize the word “experiments” -> Experiments.

Line 123 to 124 - Is there any notable variation based on the region of collection?

Line 124 - Age, shell integrity or any other physiological factors can have any influence for the dissolution rates in the shells used in these trials?

Line 129 to 131 - What interactions between the factors were noticed?

Line 133 - Capitalize “Effects of periostracum cover” -> Effects of Periostracum Cover.

Line 134 - Change “rate” to “rates”.

Line 136 to 137 - How can the seasonal variations influence the durability of periostracum across in different habitats?

Line 154 to 155 - How the factors (temperature, salinity) interfere with accurate alkalinity measurements?

Line 163 to 164 - What effects can appear from any slight chemical gradients formation?

Line 164 - Can change “establishment of any chemical” to “formation of chemical gradients”.

Line 175 to 176 - Why these pH levels were chosen? Do they correlate to natural conditions?

Line 182 to 183 - How shifts can impact different species across global environments?

Control of seawater pH

Line 209 - Capitalize “seawater” -> Seawater.

Line 210 - Why these two techniques were chosen?

Line 212 to 213 - How the variation of CO2 concentration in natural environments can affect the carbonic acid equilibrium?

Fields assays of periostracum cover

Line 229 to 230 - The type of habitat (sheltered, exposed microhabitats) will affect the periostracum coverage in natural mussel populations? How?

Statistical analysis

Line 247 - How the methods used to normalize dissolution rates will influence the outcomes?

Line 255 - Add coma after effect.

Line 257 - Change “residual” to “residuals”.

Line 274 - Remove “a” before “Welch’s”.

Results

Periostracum cover and shell loss

Line 287 - Capitalize “cover and shell loss” -> Periostracum Cover and Shell Loss.

Discussion

Line 398 to 399 - How the loss of periostracum in areas with more sun activity interact with their ability to survive? Since the ocean temperature is increasing.

Line 420 - Add coma after degradation.

Line 445 to 446 - How the increased exposure will impact the ability to adapt?

**Do you want your identity to be public for this peer review?** For information about this choice, including consent withdrawal, please see our Privacy Policy

Reviewer #1: **Yes: ** Taewon Kim

Reviewer #2: No

---

## [Author Response · Author response to Decision Letter 1]

9 Apr 2025

Editorial remarks:

Thank you for the clarification on formatting requirements; we apologize for the initial oversight and appreciate the links to pages for formatting instructions. After review, we have amended the original formatting to follow all instructions.

In your Methods section, please provide additional information regarding the permits you obtained for the work. Please ensure you have included the full name of the authority that approved the field site access and, if no permits were required, a brief statement explaining why.

We have included additional information in the Methods section describing the permit used for mussel collections.

Please remove any funding-related text from the manuscript and let us know how you would like to update your Funding Statement. Currently, your Funding Statement reads as follows: “Award 1: National Science Foundation (NSF); grant # OCE-2129942 (BG); https://www.nsf.gov/awardsearch/showAward?AWD_ID=2129942; the funders did not play any role in design, data collection/analysis, preparation of manuscript or publication decisions”. Please include your amended statements within your cover letter; we will change the online submission form on your behalf.

We have removed any reference of funding from the acknowledgments section. Additional funding that was not originally included in the portal declaration, but listed in-text, has been noted in the letter above.

Thank you for uploading your study's underlying data set. Unfortunately, the repository you have noted in your Data Availability statement does not qualify as an acceptable data repository according to PLOS's standards. At this time, please upload the minimal data set necessary to replicate your study's findings to a stable, public repository (such as figshare or Dryad) and provide us with the relevant URLs, DOIs, or accession numbers that may be used to access these data.

We apologize for not making the data publicly available at the time of review. The links included in the submission provided temporary access while our datasets were under review with the scientific data platform, BCO DMO. We have attempted to submit our dataset to Dryad after receiving editorial feedback, but were informed that our data could not be published twice, as BCO DMO has now assigned DOIs to each dataset. We then reached out to PLOS One directly and was advised that “the criteria for an acceptable repository includes perpetual access/storage, unambiguous licensing options, and (most importantly), the capacity to generate a DOI associated with the repository instance containing scientific data. If the BCO DMO repository generates a DOI and adheres to the same standards as other major repositories, then it should be regarded as an acceptable mechanism for hosting and providing publicly available data.”. We have included the published DOIs and URLs for the datasets at BCO DMO below, as this platform is reputable within the scientific community for open-source data, in hopes that the original rejection was due to the datasets being under review and not yet published.

Dataset Title: Lab incubations of mussels (Mytilus californianus) examining the influence of periostracum cover and pH on external shell dissolution at Marshall Gulch Beach, CA from August 2021 to March 2022

BCO-DMO Metadata Landing page: https://www.bco-dmo.org/dataset/935476

DOI: 10.26008/1912/bco-dmo.935476.1

Citation: Saley, A., Gaylord, B. (2024) Lab incubations of mussels (Mytilus californianus) examining the influence of periostracum cover and pH on external shell dissolution at Marshall Gulch Beach, CA from August 2021 to March 2022. Biological and Chemical Oceanography Data Management Office (BCO-DMO). (Version 1) Version Date 2024-12-28 [if applicable, indicate subset used]. doi:10.26008/1912/bco-dmo.935476.1 [access date]

Dataset Title: Lab incubations of mussels (Mytilus californianus) examining the influence of simulated abrasion of periostracum on external shell dissolution at Marshall Gulch Beach, CA from August 2021 to March 2022

BCO-DMO Metadata Landing page: https://www.bco-dmo.org/dataset/935480

DOI: 10.26008/1912/bco-dmo.935480.1

Citation: Saley, A., Gaylord, B. (2024) Lab incubations of mussels (Mytilus californianus) examining the influence of simulated abrasion of periostracum on external shell dissolution at Marshall Gulch Beach, CA from August 2021 to March 2022. Biological and Chemical Oceanography Data Management Office (BCO-DMO). (Version 1) Version Date 2024-12-28 [if applicable, indicate subset used]. doi:10.26008/1912/bco-dmo.935480.1 [access date]

Dataset Title: Field measurements of periostracum cover of mussels (Mytilus californianus) from focal population at Marshall Gulch Beach, CA in July and August 2022

BCO-DMO Metadata Landing page: https://www.bco-dmo.org/dataset/935484

DOI: 10.26008/1912/bco-dmo.935484.1

Citation: Saley, A., Gaylord, B. (2024) Field measurements of periostracum cover of mussels (Mytilus californianus) from focal population at Marshall Gulch Beach, CA in July and August 2022. Biological and Chemical Oceanography Data Management Office (BCO-DMO). (Version 1) Version Date 2024-12-26 [if applicable, indicate subset used]. doi:10.26008/1912/bco-dmo.935484.1 [access date]

Please include captions for your Supporting Information files at the end of your manuscript, and update any in-text citations to match accordingly.

Per formatting requirements mentioned above, we have included a sub-header for Supporting information content at the end of the manuscript and have updated in-text citation formatting of supplemental material.

Award 1: National Science Foundation; grant # OCE-2129942; https://www.nsf.gov/awardsearch/showAward?AWD_ID=2129942; the funders did not play any role in design, data collection/analysis, preparation of manuscript or publication decisions

Award 2: National Science Foundation; Graduate Research Fellowship Program; https://www.nsfgrfp.org/; the funders did not play any role in design, data collection/analysis, preparation of manuscript or publication decisions

Award 3: Conchologists of America Grant in Malacology; https://conchologistsofamerica.org/; the funders did not play any role in design, data collection/analysis, preparation of manuscript or publication decisions

Award 4: Russell J. and Dorothy S. Bilinski Fellowship; https://marinescience.ucdavis.edu/academics/bilinski-fellowships; the funders did not play any role in design, data collection/analysis, preparation of manuscript or publication decisions

Reviewer #1:

In this study, the authors provided evidence that the periostracum helps mussels protect against shell dissolution. I find this paper to be an interesting addition to the understanding of the protective abilities of bivalves against ocean acidification and believe it is publishable in PLOS ONE. However, the paper requires revision due to several uncertainties that need clarification before publication. Specifically, it was difficult to determine how the periostracum cover influenced the dissolution rate. I would recommend including sample images of the periostracum and the dissolved surfaces of mussels.

We are grateful to Reviewer #1 for their positive remarks concerning the value of our study. We concur that more detail regarding imaging the periostracum would be helpful, and now include an additional figure that more clearly demonstrates the contrast between the periostracum and the underlying shell in this species (new Fig. 1, lines 168-172). We also include more language describing ways in which the periostracum has been believed to influence shell dissolution. Although these concepts were largely untested prior to our study, they outline general thinking on the topic (Lines 472-474, 486-489)

Which of the two valves was used? Depending on the species, the two valves may not be symmetrical.

Mytilus mussels are bilaterally symmetrical, such that the left and right valves have near-equivalent size and shape. We now state this information for the reader (Lines 158-160).

Is it possible to see the periostracum with the naked eye? What criteria were used to identify the periostracum? Please provide additional details in the supplementary material. It was difficult to understand how varying percentages of periostracum cover influenced the dissolution rate. Could you provide sample images for each end of the spectrum? This would make it much clearer.

We thank the reviewer for posing these three related questions. In Mytilus californianus, the periostracum is visible to the naked eye, appearing as a dark brown to black covering over a lighter grey to light bluish purple that characterizes the underlying shell. These color differences are reinforced by discrete edges to the periostracum in locations where it is discontinuous. We now clarify these visible criteria, and add a new figure that shows images before and after delineating periostracum from the underlying shell, including a range of coverage percentages (new Fig. 1, lines 168-172).

Why do mussels abraded with fine sand show a negative dissolution rate (lower than controls)?

In Fig 5 (originally Fig 4), dissolution rates are plotted relative to a periostracum-free control shell, where y = 0 indicates an equivalent dissolution rate to controls. Thus, negative dissolution rates indicate that when shells were abraded with fine grit, the measured shell loss was less than that of unabraded control shells. We now streamline the pertinent phrasing in the text and highlight this point further in the figure caption (lines 424-426). In terms of the mechanistic basis for this pattern, we suspect it may be due to reduced surface roughness (and thus less total surface area at the microscopic level that could contribute to dissolution) in shells abraded with the fine sand.

Is there a specific reason for not using SEM? SEM offers much higher resolution and could provide more precise results. Why wasn’t the periostracum thickness measured, even though it is important?

We appreciate the reviewer inquiring about using scanning electron microscopy or other high-resolution techniques to measure the periostracum and its thickness. Unfortunately, although we agree with the reviewer that such methods would have been appropriate and potentially informative, SEM was not readily available to us. We emphasize that we are confident that our digital imaging technique is robust and adequate to characterize the broad scope of variation in periostracum cover observed in our mussels. We do now note in the text that future research could use higher resolution instrumentation to follow up on our work (lines 533-536).

Reviewer #2:

This study has the objective to investigate the impact of periostracum coverage on the ability of California mussels to resist shell dissolution under a diverse environment stressors, especially ocean acidification, where the authors explored how shell size, age and environmental conditions (such as food availability and wave exposure) can interact with the periostracum to influence dissolution rates. The paper is very well written, well-organized and insightful, providing clear explanations of the experimental design and results.

We are grateful to the reviewer for their supportive remarks.

You can change “possibility” to “hypothesis”. You can change “Because” to “Since”. If you have enough space, you should specify the meaning of “rougher surface texture” and its effects on dissolution.

We agree with the word change suggestions and have amended accordingly (lines 31, 34). We also appreciate the request to clarify surface texture and have added text to explain how rougher textures may link to higher dissolution rates (lines 39-43).

How the desiccation at low tide can contribute to damage?

We thank the reviewer for encouraging us to explain the connection between low tide and periostracum damage. We now clarify better that desiccation and temperature cycling at low tide could weaken the periostracum’s attachment to the underlying shell, which may result in higher amounts of detachment (lines 43-46).

Add how mussels modify local physical factors and its impacts (water flow, sediment stability, etc).

We are grateful for the request to add more details as to how mussels modify local factors and have amended the text to include the specific factors explored in the referenced studies (lines 58-60).

On mechanical and chemical armour, does it mean periostracum will prevent shell thinning? Or does it neutralize acidic conditions?

We thank the reviewer for highlighting the need to better define ‘armour’ from the referenced study. The periostracum acts as both a mechanical and chemical barrier, protecting the shell from seawater conditions that may encourage dissolution. We now clarify this meaning for the reader (lines 68-70).

Why do particles attached reduce OA-associated dissolution?

We appreciate the question for highlighting the need to elaborate on how clay particles around the periostracum and shell may reduce dissolution in corrosive seawater. The referenced study discusses how clay shields protect gastropod shells in low-pH estuaries by creating a low-permeability barrier. We’ve added further clarification for the reader (lines 73-76).

What are the implications [of periostracum thickness and composition]? Are there variations in the chemical compositions among different mussels species that can affect its ability to withstand OA?

We agree with the request to expand on the implications of periostracum thickness and composition in dictating the degree of shell corrosion in low pH seawater. We now clarify more explicitly how these traits influence the severity of corrosion by highlighting their potential to mediate contact between seawater and the underlying shell (lines 78-79).

You can change “may propagate” to “may disrupt”.

We appreciate the suggestion; however, we believe the change would alter the intended meaning. Instead, we have revised the text to emphasize the significance of understanding how individual responses cascade to influence relationships among species (lines 87).

Change “not living tissue” to “non-living tissue”.

We appreciate this suggestion and have amended accordingly (line 88).

What are the molecular mechanisms in response to acidification? Microbial activity impacts or influences the degradation of periostracum?

We agree that additional molecular studies and explorations of microbial effects would be valuable, but such efforts are outside the realm of our study.

How periostracum degradation varies across different intertidal zones/ecosystems? Does UV radiation have any impact/role on structural integrity? What are the long-term effects of tidal exposure?

We appreciate the note on variability in mechanisms driving degradation in intertidal habitats and the suggestion to include UV radiation as a factor influencing periostracum integrity. Additionally, we appreciate the request for clarification on effects of tidal exposure. We now expand on these connections in line 99.

Change “undertook” to “conduct”.

We have amended the phrasing as suggested (line 105).

Is it possible to identify the specific conditions where the periostracum loss accelerates?

We appreciate the question; however, due to our experimental design testing only three discrete pH conditions, we are unable to discern whether dissolution accelerates in a non-linear fashion outside our tested range. We now highlight this point more clearly in the discussion section to improve transparency of experiment limitations (lines 494, 498).

Predation or microbial actions are most effective in removing periostracum?

We agree with the reviewer that predation and microbial activity are likely effective agents in the r

---

## [Decision Letter · Decision Letter 1]

Dear Dr. Saley,

Thank you for submitting your manuscript to PLOS ONE. After careful consideration, we feel that it has merit but does not fully meet PLOS ONE’s publication criteria as it currently stands. Therefore, we invite you to submit a revised version of the manuscript that addresses the points raised during the review process.

We look forward to receiving your revised manuscript.

Kind regards,

Michael Schubert

Academic Editor

PLOS ONE

Journal Requirements:

Reviewers' comments:

Reviewer's Responses to Questions

**Comments to the Author**

Reviewer #2: All comments have been addressed

Reviewer #3: All comments have been addressed

2. Is the manuscript technically sound, and do the data support the conclusions?

Reviewer #2: Yes

Reviewer #3: Yes

3. Has the statistical analysis been performed appropriately and rigorously?

Reviewer #2: Yes

Reviewer #3: Yes

4. Have the authors made all data underlying the findings in their manuscript fully available?

Reviewer #2: Yes

Reviewer #3: Yes

5. Is the manuscript presented in an intelligible fashion and written in standard English?

Reviewer #2: Yes

Reviewer #3: Yes

Reviewer #2: The corrections and explanations provided were well-written and cohesive. The additional information added enhanced the context and improved the readability of the paper, leading to a better understanding. However, it would be beneficial to include some further details, as suggested below, so minor revision are needed. Overall, excellent work and writing, the revisions made were well-done.

Abstract

Really good explanations were added, adding more context made the abstract clearer.

Line 68 to 70 - Is it possible to add whether the periostracum has any chemical capacity against the acidic conditions?

Line 73 to 76 - The explanation added is relevant and good, but the phrasing could be slightly better to add clarity.

Line 78 to 79 - Even though the information added is helpful, you could clarify the composition of the periostracum varies among different mussels species in this part.

Line 107 to 109; 568-569 - Adding the specific physical agents found in intertidal habitats that you explored would be good in the first part. For the discussion part, maybe adding further examples of how predation and microbial activity interact with physical agents in the removal of periostracum would be good too.

Line 190 to 195 - The link between temperature, salinity and alkalinity measurement could be made clearer.

Line 498 to 500 - Good adding.

Line 513 to 515 - Clarifying if other physiological factors have been observed to influence the dissolution rates in this part would be good.

Reviewer #3: (No Response)

**Do you want your identity to be public for this peer review?** For information about this choice, including consent withdrawal, please see our Privacy Policy

Reviewer #2: No

Reviewer #3: No

---

## [Author Response · Author response to Decision Letter 2]

9 Jun 2025

Response to Reviewers

We thank the two reviewers for their careful evaluation of our manuscript. We have clarified the chemical protective role of the periostracum, better defined the physical and biological agents involved in periostracum removal, and expanded explanations on the importance of temperature and salinity measurements for accurate seawater carbonate analysis. These revisions have enhanced the clarity of our study, and we appreciate the reviewer’s insights.

Below, we respond (in italicized text) to each comment in sequence, referencing the relevant line number of the revised manuscript.

Sincerely,

Dr. Alisha Saley

Reviewer #2:

The corrections and explanations provided were well-written and cohesive. The additional information added enhanced the context and improved the readability of the paper, leading to a better understanding. However, it would be beneficial to include some further details, as suggested below, so minor revision are needed. Overall, excellent work and writing, the revisions made were well-done.

Really good explanations were added, adding more context made the abstract clearer.

We are pleased that Reviewer #2 supports the corrections made to improve the explanations, context, and readability of our paper. We appreciate the kind remarks regarding our efforts.

Is it possible to add whether the periostracum has any chemical capacity against the acidic conditions?

We thank the reviewer for encouraging us to clarify our wording regarding the potential for the periostracum to chemically shield the underlying shell from corrosive seawater conditions. We now more clearly explain that the ‘chemical capacity against acidic conditions’ refers to the hydrophobic nature and low porosity of the periostracum’s layered structure—i.e., the chemical properties of the periostracum material itself, rather than chemical reactions between the periostracum and seawater—as a means of protecting the underlying shell (lines 67–68, 74). While we do not speculate on possible chemical interactions between the periostracum and seawater, as this lies beyond the scope of our study, we emphasize the value of future research exploring this line of inquiry.

The explanation added is relevant and good, but the phrasing could be slightly better to add clarity (Lines 73-76). Even though the information added is helpful, you could clarify the composition of the periostracum varies among different mussels species in this part (Lines 78-79).

We appreciate the reviewer’s request to clarify that periostracum thickness, porosity, and organic composition are characteristics that may influence the extent of shell corrosion. In our revised explanation, we also note that these characteristics can vary among mussel species (lines 75–77).

Adding the specific physical agents found in intertidal habitats that you explored would be good in the first part. For the discussion part, maybe adding further examples of how predation and microbial activity interact with physical agents in the removal of periostracum would be good too (Lines 107-109; 568-569).

We thank the reviewer for highlighting the need to better define the physical agents examined in our study. We have added further clarification for the reader (lines 109–110, 114–115). Additionally, we included text clarifying that predation and parasitic activity (i.e., biological agents of removal) may interact with, and be influenced by, physical agents in the removal of the periostracum layer (lines 495, 505–510).

The link between temperature, salinity and alkalinity measurement could be made clearer (Line 190-195).

We thank the reviewer for encouraging us to clarify the importance of measuring temperature and salinity to accurately analyze total alkalinity and subsequently calculate other parameters in the seawater carbonate system (lines 181–182). We have explained how temperature and salinity influence seawater pH and emphasized their critical role in the accurate measurement and calibration of spectrophotometric methods (lines 189–191). Additionally, we included a brief description of how alkalinity is analyzed via acid titrations (lines 202–205). Finally, we provide a more detailed explanation of the relationship between salinity and alkalinity and why precise salinity measurements are essential for calculating alkalinity from a titration curve (lines 205–211).

Good adding (Lines 498-500).

We are grateful to the reviewer for their supportive remarks.

Reviewer #3: (no response)

Editorial Remarks:

As requested by the editor, we have thoroughly reviewed our reference list for accuracy and completeness. No new references were added, and all listed references were cross-checked against the Retraction Watch Database to confirm none have been retracted. The amendments made to the reference list focus on formatting improvements, such as replacing URL links with DOIs where applicable.

---

## [Editor Report · Decision Letter 2]

Mussel periostracum protects against shell dissolution

PONE-D-24-55358R2

Dear Dr. Saley,

We’re pleased to inform you that your manuscript has been judged scientifically suitable for publication and will be formally accepted for publication once it meets all outstanding technical requirements.

Kind regards,

Michael Schubert

Academic Editor

PLOS ONE

---

## [Editor Report · Acceptance letter]

PONE-D-24-55358R2

PLOS ONE

Dear Dr. Saley,

I'm pleased to inform you that your manuscript has been deemed suitable for publication in PLOS ONE. Congratulations! Your manuscript is now being handed over to our production team.

Kind regards,

on behalf of

Dr. Michael Schubert

Academic Editor

PLOS ONE